# Molecular Dynamics of Trogocytosis and Other Contact-Dependent Cell Trafficking Mechanisms in Tumor Pathogenesis

**DOI:** 10.3390/cancers17142268

**Published:** 2025-07-08

**Authors:** Haley Q. Marcarian, Anutr Sivakoses, Alfred L. M. Bothwell

**Affiliations:** 1The University of Arizona Cancer Center, The University of Arizona, Tucson, AZ 85721, USA; hmarcarian@arizona.edu (H.Q.M.); asivakoses@arizona.edu (A.S.); 2Department of Pathology, Microbiology, and Immunology, The University of Nebraska Medical Center, Omaha, NE 68198, USA; 3Department of Immunobiology, Yale School of Medicine, New Haven, CT 06510, USA

**Keywords:** trogocytosis, entosis, cell fusion, nanotubes, cell–cell trafficking

## Abstract

During oncogenesis, cancer cells must develop complex strategies to proliferate, survive, and avoid the body’s natural defenses. One class of strategies tumor cells can use is contact-dependent transfers of cellular materials. This review highlights four important mechanisms by which cancer cells exchange and obtain cellular components from their neighbors. These processes can result in increased metastatic capacity, better evasion of immune detection, drug resistance, genomic instability, and dysregulated metabolism.

## 1. Introduction

Horizontal integration of cellular components originating from an unrelated cell is a staple biological process that is well characterized in other organisms, such as horizontal gene transfer in bacteria, first characterized in 1928 by Griffith [1]. Other model organisms utilized in laboratory organisms also undergo horizontal trafficking, such as the formation of thin, tubular trafficking networks known as cytonemes in *Drosophila melanogaster* [2]. In mammals, Chargaff and West recorded the first instance of transfer of subcellular compartments between two cells in the form of platelet-derived extracellular vesicles [3].

Cancers are dynamic diseases that can adapt to their surrounding microenvironments in order to survive and proliferate. Although it is accepted that external factors could contribute to the increase in tumor mutational burden, this accumulation only affected a small subset of cells, which would ultimately undergo clonal expansion to form a tumor. The characterization of the downstream consequences of cell–cell trafficking within the context of cancer only began as recently as 2006 [4]. Further studies have demonstrated how intercellular transfer of DNA, RNA, and protein alter the interplay between tumor and other resident tumor microenvironment (TME) cells [5,6,7,8]. This review compiles the molecular mechanisms that drive four unconventional near-cell trafficking methods and the outcomes of each method in reshaping tumor plasticity, dampening anti-tumor immunity, and increasing the tumor mutational burden (Figure 1).

## 2. An Overview of Contact-Dependent Cell–Cell Trafficking Methods

### 2.1. Overview of Trogocytosis

#### 2.1.1. A Historical Overview of the Biological Functions of Trogocytosis in Physiology and Disease

The contact-dependent transfer of plasma membrane from a donor cell to a recipient cell is known as trogocytosis, and was first described by Cone, Sprent, and Marchalonis in 1972 [9,10]. Trogocytic interactions were first characterized as a unidirectional transfer of membrane-bound proteins and lipids from an antigen presentation cell to a lymphocyte and was demonstrated to be a critical process in CD4+ and CD8+ T cell recognition of antigens [11,12]. Other essential physiological functions have been implicated in trogocytosis, such as the spermicidal function of vaginal neutrophils [13]. Neuronal circuit maturation is also dependent on microglia trogocytosis between presynaptic boutons and axons on nerve cells [14]. In contrast, pathogens can also harness trogocytosis to avoid selective pressure originating from the host environment [15,16]. *E. histolytica parasites* trogocytose membrane fragments of human cells and lead to a sharp influx of Ca^2+^ into the human cells, resulting in lytic death [15].

#### 2.1.2. Trogocytosis Within the Tumor-Immune Microenvironment

In cancer, trogocytosis alters the transcriptome of cells within the tumor-immune microenvironment (TIME). Trogocytosis between tumor and immune cells can occur in both directions and leads to decreased immune cell sensitization through the transfer of functional immune checkpoint proteins, increased lymphocyte fratricide and “self” peptides [17,18,19,20,21]. In contrast, tumor-immune cell trogocytosis can also enhance the cytotoxic metabolism of effector immune cells, leading to enhanced tumor clearance [22]. Recently, the transfer of genomic DNA was also implicated in trogocytic interactions between tumor and immune cells, although it is currently unknown whether this transfer is a direct consequence of trogocytosis or driven by different interactions that share conserved pathways with [17,20,21].

### 2.2. Overview of Entosis

The term “entosis” was first coined by Overholtzer et al. in 2007, when they described this process as a nonapoptotic cell death process that is initiated by loss of adherence to the extracellular matrix [23]. This phenomenon was unique, as it involved the invasion of one living cell into another, resulting in the formation of a “cell-in-cell” structure (CIC) [23]. These CIC structures can refer to one or multiple individual cells invading a single host [24]. The engulfed cell, commonly referred to as the “loser”, can encounter one of several fates. Death by lysosomal acidification of the entotic vacuole containing the “loser” cell is driven by the LC3-mediated recruitment of autophagy-related proteins and is the most frequently observed outcome [25]. However, it is possible for engulfed cells to escape their host or even undergo cell division while still entrapped [23]. If the internal cell divides, the resulting daughter cells will either die, escape, or simply remain internalized in their host for an unknown amount of time [23]. In the context of cancer, the significance of entosis is multifaceted, as it has a role in both pro- and anti-tumor processes. On the one hand, entosis may limit tumor metastasis by eliminating cancer cells that break free from the surrounding extracellular matrix [23,25,26]. However, researchers have observed a correlation between the number of entotic cells and severity of disease across multiple types of cancer [27,28].

### 2.3. Overview of Cell Fusion

Cell fusion refers to the process by which two cells in contact with each other merge surface membranes and combine into a single daughter cell. In normal biology, there are several processes involving cell fusion that occur in mammals, including the fusion of sperm and egg, the fusion of myoblasts into myotubes during muscle development, the fusion of macrophages to form multinucleated giant cells, and the fusion of monocytic cells to form osteoclasts [29,30,31,32]. In cancer, cell fusion is thought to be a relatively rare event, with researchers estimating that 0.0066 and 6.5% of cancer cells undergo this process in vitro [33]. Although rare, cell fusion is believed to have a significant role in shaping tumor metastasis, drug resistance, and heterogeneity. Researchers have shown that cell fusion in cancer can lead to parasexual recombination of genetically distinct cancer cells, leading to intratumoral heterogeneity [34].

### 2.4. Overview of Tunneling Nanotubes

Tunneling nanotubes (TNTs) and tumor microtubes (TMs) are three-dimensional, hollow, and tubular structures that connect two cells and are similar to cytonemes found in *D. melanogaster* [35]. They are classified based on tube length and thickness. TNTs and TMs allow for the transfer of subcellular components and are most commonly observed in neurons, astrocytes, and glial cells, being essential in the maintenance of the neuronal networks of the central nervous system in normal physiology [36]. The main structural components of TNTs and TMs are filamentous actin (F-actin) fibers coated with the cell membrane. TNTs and TMs can traffic functional pieces of subcellular components, including cell membrane fragments, mitochondrial DNA (mtDNA), and protein [6,7,37]. In solid cancers, structures such as TNTs and TMs were first characterized in 2004 by Rustom et al. in a rat adrenal tumor and have since been discovered to be utilized by cancer cells to alter gene and protein expression in the TME of human breast, ovarian, cervical, lung, and pancreatic cancers [38]. The formation of TNTs is not limited to two cancer cells and has been shown to form between a tumor cell and a T cell [7]. The formation of TNTs is also observed to lead to the increased expression of epithelial-to-mesenchymal transition (EMT)-related proteins in cancer cells [39]. Taken together, TNTs promote tumor development and progression through the shuttling of functional DNA, organelles, and oncogenic proteins in order to elevate tumor cell respiration and mutational burden.

## 3. Mechanisms of Cell–Cell Trafficking

### 3.1. Mechanisms of Trogocytosis

#### 3.1.1. Solid Tumor Trogocytosis

Very little is known about the mechanisms that drive trogocytosis. Within the context of cancer, most studies investigated the immunomodulatory effects of trogocytosis in tumor-immune cell synapse rather than the physiological pathways that induce trogocytosis. Trafficking via trogocytosis occurs in both directions, where the tumor cell is able to act as both a donor and a recipient cell [17,18,20,21,40,41]. However, it is currently unknown whether this “swap” occurs concomitantly or if there are two separate trogocytic interactions. The initiation of trogocytosis is dependent on the physical contact between two cells [12]. It does not occur between two cell populations when separated by a barrier such as a transwell in in vitro cocultures [18,20,21].

Trogocytosis is also associated with elevated or aberrant phosphatidylinositol 3-kinase (PI3K) activity [42]. Overactivation of PI3K results in hyperproliferation in cancer cells, increasing cancer cell growth rate and apoptotic resistance. Studies assessing the immunomodulatory effects of trogocytosis have shown that Wortmannin and Latrunculin A/B suppress trogocytosis by inhibiting F-actin synthesis [42]. However, the conclusions drawn from studies using these inhibitors to prevent tumor cell acquisition of lymphocyte protein via trogocytosis are controversial, as expression of PI3K is necessary for cell maintenance and survival (Figure 2).

#### 3.1.2. Liquid Tumor Trogocytosis

In contrast, trogocytic interactions between leukemia cells, a bone marrow-derived liquid tumor, and natural killer (NK cells) were found to be driven by SLAM-receptor signaling [18]. SLAM receptors are type I glycoproteins that are expressed on a subset of immune cells, including activated T cells, B cells, macrophages, and dendritic cells [43]. However, it is still unknown what specific downstream effectors of SLAM receptors drive trogocytosis. If the mechanism is similar to the immune cell trogocytosis of solid tumor cells, NK cell-driven trogocytosis may involve SLAM receptor-mediated recruitment of PAK-interacting protein (β-PIX), a guanine nucleotide exchange factor that activates Rho/ROCK-mediated F-actin polymerization [44]. This hypothesis would be consistent with the findings that treatment with latrunculin A also impeded Chimeric Antigen Receptor T cell (CAR-T) trogocytosis of multiple myeloma cells (Figure 2) [45].

### 3.2. Mechanisms of Entosis

#### 3.2.1. ROCK Signaling Pathway

Entosis is a complicated process that is primarily driven by the internalized cell, rather than the engulfing cell, which distinguishes it from other processes, such as phagocytosis (Figure 3). Mechanotransduction and differences in the stiffness of the membranes of the host and invading cell are critical for entosis to occur. Differences in the activation of myosin II specifically has been shown to promote entosis, as cells with higher myosin II activity are more likely to invade their neighbors [46]. This invasion is primarily driven by ROCK signaling [23]. More specifically, Cdc42, Rho (RhoA, RhoB, and RhoC), and Rac (Rac1, Rac2 and Rac3) are the most critical small GTPase members of the Rho family when it comes to driving entosis [47,48]. Activation of the ROCK signaling pathway leads to a higher concentration of active Rho in the distal end of the invading cell, opposite its point of contact with the host. The resulting mechanical tension promotes cellular invasion and subsequent entosis [48,49].

#### 3.2.2. External Drivers of Entosis

Loss of cell adhesion to the extracellular matrix has been associated with entosis since its initial discovery [23]. In normal tissues, cells that detach from the matrix typically undergo a programmed cell death pathway, such as anoikis or apoptosis. This likely explains why the most common place to find CIC structures is in the fluid exudates of cancer patients (urine, bile, ascites, pleural fluid, etc.) [50]. However, cancer cells can overcome this by altering integrin expression, upregulating pro-survival pathways such as PI3K/Akt, and altering their metabolism [51]. After detaching from the basement membrane, free floating cells must form adherens junctions with the host cell for entosis to begin [23,50,52,53]. After adherens junctions are established, an imbalance of actin-myosin contractions between the cells promotes the invasion of the internalizing cell, and entosis occurs [46,48].

Another external factor that can influence entosis is nutrient deprivation. Researchers have shown that under these conditions, the engulfing cell is able to harvest nutrients, such as amino acids, from the invader, ultimately supporting the survival of the engulfer [54]. Glucose deprivation appears to have the most profound effect on rates of entosis. For example, Hamann et al. have shown that increased activity of the starvation-induced kinase AMPK can skew a cell towards an “invader” phenotype [55]. This may lead to increased competition between cancer cells, such that engulfing cells may be able to promote their own survival through scavenging nutrients from invading cells under starvation conditions.

Ultraviolet radiation (UV) is also capable of inducing entosis. In 2021, Chen et al. showed that when exposed to a high dose of UV (100 J/m^2^), 35% of MCF7 human breast cancer cells underwent entosis [56]. This study demonstrated that UV-induced entosis was regulated by c-Jun N-terminal kinase (JNK) and p38 stress-activated kinases in the internalized cell [56]. Ingesting other cells seems to provide a survival advantage to the host cell when challenged by UV radiation, indicating that nutrients scavenged from an internalized cell can provide some protection from various stressors to the host [56]. It remains unknown whether non-transformed cells in vivo exhibit a similar increase in entosis in response to UV radiation. This is an important area of investigation, as the UV-induced entosis of normal cells may contribute to early cancer cell transformation and genomic instability.

### 3.3. Mechanisms of Cell Fusion

The process of cell fusion can be broken down into five major steps: cell priming, chemotaxis, adhesion, fusion, and post-fusion (Figure 4) [57]. In general, these steps are critical in order for the fusing cells to overcome the barrier posed by the integrity of the phospholipid bilayer membranes of the cells’ surfaces. Cell fusion is important for several biological processes in normal physiology, including fertilization, muscle development, skeletal remodeling through the generation of multinucleated osteoclasts, and placentation [58,59]. However, when this process becomes dysregulated, it can lead to the development of cancer [60,61].

#### 3.3.1. Priming

To undergo cell fusion, cells must first enter a priming phase that prepares the machinery necessary for successful fusion to occur. This involves making modifications to the lipid composition of the cell membrane by translocating the inner-leaflet lipids, including phosphatidylserine, to the cell surface [57]. During fertilization, exposed phosphatidylserine (PS), localized to the anterior acrosomal region of the sperm, is thought to be critical for membrane destabilization, thereby enabling penetration of the zona pellucida [57,62]. Next, the architecture of the cell membrane is altered to make it more fluid through the depletion of cholesterol and filipin in an albumin-dependent manner [63]. While the significance of PS in driving cancer cell fusion is not well understood, early studies have shown that it is likely necessary [33,64,65].

#### 3.3.2. Chemotaxis

Very little is known about the role of chemotaxis in driving cancer cell fusion; however, sperm, myoblasts, macrophages, and other cells rely on the secretion of chemoattractants to find competent fusion partners [57]. There are many cytokines and chemokines that can facilitate cell fusion; however, the most relevant to cancer are platelet-derived growth factor (PDGF), tumor necrosis factor alpha (TNF-α), epidermal growth factor (EGF), and interleukin-4 (IL-4) [66,67,68,69,70,71]. Specifically, it is believed that chemokine (C-C motif) ligand 2 (CCL-2) plays a major role in recruiting cells such as M2 macrophages to the tumor site [72,73]. M2 macrophages are a type of pro-tumor-associated macrophage and have previously been shown to undergo cell fusion with tumor cells [74]. Cell fusion can also result in a more metastatic and mobile phenotype of the cancer cell. A study authored by Xia et al. demonstrated transcriptomic changes to chemokine signaling pathways post-cell fusion, resulting in a more migratory cancer cell phenotype [75] Cell fusion can also result in tumor cells retaining expression of the hematopoietic marker CD45 through multiple lineages after undergoing cell fusion with hematopoietic cells [76].

#### 3.3.3. Adherence

For cells to undergo cell fusion, they must be in physical contact with one another. There are several cell surface adhesion proteins that promote adhesion and recognition between the two cells, including cadherins, tetraspanins, integrins, and immunoglobulin super family members.

Cadherins that participate in homotypic cell–cell contact have been shown to have a role in many types of cell fusion. Researchers have shown that treatment of macrophages with anti-E-cadherin antibodies blocks the formation of multinucleated giant cells (MGCs) [77]. MGCs are formed through the fusion of macrophages and play an important role in bone remodeling and innate immunity [78]. During osteoclastogenesis, E-cadherin expression by osteoclasts can influence gene expression to promote migration and fusion [79]. While E-cadherin expression generally suppresses tumor formation in normal cells, its dysregulation has been shown to lead to carcinogenesis [80]. It is possible that dysregulation of E-cadherin can drive cancer cell fusion and malignancy; however, a direct connection between E-cadherin and cancer cell fusion has yet to be demonstrated.

Tetraspanins are cell surface molecules that can interact with other cell surface proteins such as integrins and Ig superfamily proteins. In mouse models, researchers have shown that the tetraspanins CD9 and CD81 are needed to facilitate the fusion of the egg and sperm [81,82,83]. The neutralization of CD9 in macrophages also disrupts the formation of osteoclasts through fusion [84]. In contrast, CD9 and CD81 can limit the fusion of cells in certain circumstances, as CD9- and CD81-deficient mice have increased numbers of MGCs during inflammation [85]. Tetraspanins are also known to associate with certain integrins to promote cell fusion. For example, β1 integrin is thought to encourage myoblast and myotube fusion by organizing and controlling the expression of fusion proteins including CD9 [86].

#### 3.3.4. Fusogens and Membrane Pore Formation

The most widely accepted model of cell-fusion pore formation begins with the formation of a hemifusion intermediate in which the outer lipid layer of the surface membranes of each fusing cell are the first to merge with each other. During the hemifusion intermediate stage, the outermost leaflets of opposed membranes are connected, while the inner leaflets remain intact [87]. Next, the tension in the diaphragm, generated by the inner leaflets, promotes their merging and the subsequent formation of a fusion pore [88]. After the initial pore is formed, it expands and allows the mixing of intracellular contents in a single body. Under normal conditions, the plasma membrane of cells will not spontaneously fuse together due to the high energy required to overcome the forces opposing cell fusion. However, a class of proteins known as “fusogens” has been shown to lower the energy barrier by altering the lipid bilayer and to facilitate fusion [89,90]. Fusogens play an important role in mediating viral entry into host cells [88]. They are typically homo- or hetero-oligomeric complexes composed of transmembrane glycoproteins that can be activated by changes in pH [8,91]. Identifying fusogens in mammalian cells has remained challenging, with the exception of Myomaker (Mymk) and Myomerger (Mymg). When they are expressed endogenously, these two fusogens have been shown to drive the fusion of activated muscle cells [92,93,94,95]. To date, however, Mymk and Mymg have yet to be definitively implicated in driving cell fusion in any form of cancer.

Syncytin-1, which is a membrane glycoprotein encoded by the human endogenous retrovirus element HER-W, is similar in structure to HIV Env fusogen and is believed to play a role in aberrant cancer cell fusion [57]. In normal cells, syncytin-1 and its homolog, syncytin-2, drive the formation of the placental syncytiotrophoblast [96]. However, aberrant expression of syncytin-1 has been implicated in promoting cell fusion in endometrial carcinoma and it may have a tumorigenic role in non-small cell lung cancer (NSCLC), breast cancer, neuroblastoma, and testicular cancer [97,98,99,100]. Syncytin-1 promotes cell fusion by binding to the receptors ASCT1 or ASCT2, which are also overexpressed in some cancers [101,102].

#### 3.3.5. Post-Fusion Recovery

After fusion, cells must undergo a period of resetting to either prevent themselves from undergoing another fusion event or prepare to fuse again [31]. One of the first problems the new cell must address is the surplus plasma membrane, which can result in suboptimal cell surface tension [57]. Myoferlin is a member of the ferlin family of proteins, and it is highly expressed in myoblasts undergoing a period of resetting [103]. Myoferlin is responsible for promoting the endocytic recycling of excess plasma membrane in an EHD2-dependent manner [104]. The anti-apoptosis proteins Bcl-2 and c-Flip are also believed to be upregulated in recently fused cells to prevent unwanted cell death [105,106].

### 3.4. Mechanisms of Tunneling Nanotubes and Tumor Microtubes

#### 3.4.1. Differences Between TNTs and TMs

Cancer cells form TNTs and TMs with cell types, including immune cells and other cancer cells. TMs and TNTs are similar in function and are differentiated by their size and width. TMs are typically longer (500 µM) and have a larger bore size (1–2 µM) compared to TNTs (100 µM length and less than 1 µM width) [39,107,108]. Additionally, branching and multiple connections are more frequently observed in TNTs. TMs and TNTs are both sheathed in the host cell membrane [108]. Due to their thinness, TNTs have also been observed to be more fragile compared to TMs, which can survive multiple washes and buffer replacements during immunofluorescence labeling [7,109]. Both have been observed to form between cancer cells in vitro and in vivo [6,36,110]. TMs are often visualized using brightfield or phase contrast microscopy, whereas TNTs typically require higher-resolution imaging techniques, such as scanning electron microscopy (SEM) [7]. However, TMs and TNTs are functionally similar and are both utilized by tumor cells to connect with other cells for the purposes of trafficking functional fragments of protein, nucleic acids, and subcellular structures.

#### 3.4.2. The Molecular Mechanisms of TNT and TM Formation and Structure

In normal physiology, the expression of p53 is essential for TNT formation between astrocytes [111]. However, a hallmark of many tumors is the loss of p53 function, signifying that tumors may rely on an alternative mechanism to drive the formation of TNTs/TMs [112]. Additionally, it is also currently unknown whether the mechanisms that drive TMs and TNT formation are conserved. The formation of both TNTs and TMs in tumors is driven by actin polymerization [38,111,113,114]. Treatment with drugs that promote F-actin depolymerization, such as Latrunculin B, also prevents TNT and TM formation and promote the collapse of preexisting TNTs and TMs [109].

Tumor necrosis factor α-induced protein (TNFAIP2) has been implicated in TNT formation through its interaction with RalA, a GTPase belonging to the Ras superfamily of proteins [115,116,117]. RalA interaction with guanosine-5’-triphosphate (GTP) is catalyzed by guanine exchange factors (GEFs) [118,119]. The RalA/GTP complex then binds to RalA Binding Protein 1 (RALBP1), activating Rac1/RhoA-mediated actin polymerization [120]. The exocyst complex, an octomeric protein complex consisting of Exo70, Exo80, Sec3, Sec5, Sec6, Sec8, Sec10, and Sec15, has also been implicated in the formation of nanotubes [115,121]. It is believed that RhoA-mediated F-actin synthesis downstream of the exocyst complex, which itself is a downstream effector of TNFAIP2-RalA interactions, can also induce nanotube formation. Direct inhibition of TNFAIP2 via RNA interference (RNAi) and inhibition of TNFAIP2 and Ral interactions led to decreased de novo TNT synthesis [122,123].

Connections formed between a TNT or TM and a cell rely on the expression of connexin 43 (Cx43), a protein that forms gap junctions [124,125]. Gap junctions formed as a result of Cx43 expression facilitate the movement of cellular components into and out of the TNTs and TMs [126]. Cx43 may also further induce TNT and TM formation, as its expression also stimulates PI3K signaling, leading to further downstream actin polymerization. Beyond p53 and PI3K, overexpression of the mutant KRAS^G12D/G13D^ oncogene has led to increased formation of TNTs and TMs [6]. These signaling pathways also stimulate TNFAIP2 function, suggesting that there is no one master regulator of TNT and TM formation.

To summarize, many of the mechanisms that drive TNT and TM formation are regulated by de novo F-actin synthesis, similar to phagocytosis and entosis (Figure 5) [23,24,127]. Although TNT and TM development does not require physical contact between two cells, they are considered to be very fragile and cannot be formed through physical barriers or pores [7]. For example, Saha et al. in 2022 investigated the transfer of mtDNA from T cells and breast tumor cells and established that TNTs failed to form between T cells and cancer cells when separated by a transwell [7]. This study also demonstrated the fragility of the nanotubes, as evidenced by images captured via field emission scanning electron microscopy (FESEM) [7]. TNTs and TMs are more prone to breakage, resulting in incomplete transfer of components between cells compared to shorter nanotubes [7,39,107,108]. The current hypothesis of the molecular mechanisms that drive TNT formation is that cooperative interaction between TNFAIP2 and RalA drives the recruitment of the exocyst complex, which in turn elevates de novo synthesis of long, thin F-actin filopodia connecting two or more cells. Although it is unknown whether TNFAIP2 also drives the formation of TMs, the structural and functional similarities between TMs and TNTs suggest that the pathways that drive their formation are conserved.

### 3.5. Similarities Between the Mechanisms That Drive Phagocytic Cup Formation and Other Cell–Cell Trafficking Processes

#### 3.5.1. Recognition of Target Cells and Particles Flagged for Phagocytosis

Phagocytosis is a programmed cellular function that results in the engulfment and complete lysis of a cell or particle. Although phagocytosis is not considered a method of cell–cell trafficking, there are similarities between the mechanisms that drive phagocytosis and the cell–cell trafficking methods previously highlighted.

Professional phagocytes mainly consist of innate immune cells such as macrophages, neutrophils, and monocytes [128]. They are recruited to the site of trauma over short- and long-range through chemoattractants, including complement activators C3a and C5a, which are secreted by tissue resident cells [129]. Professional phagocytes express unique “eat-me” or phagocytic receptors that recognize pathogen-associated molecular patterns (PAMPs) and/or opsonin receptors that detect antibodies, extracellular proteins, and other “tags” that signal the target cell for engulfment. Professional phagocytes also express “don’t-eat-me” receptors, such as signal regulatory protein α (SIRPα), which can attenuate phagocytic activity upon recognition of ligands, such as CD47 [130,131].

#### 3.5.2. Formation of the Phagocytic Cup Scaffolding

The process of phagocytosis is formally initiated when phagocytic or opsonin receptor bind to an “eat-me” ligand [132]. A commonly recognized ligand is PS, a typical marker for cells undergoing apoptosis [133,134]. Upon ligand recognition, depolymerization of the phagocyte’s cortical actin cytoskeleton occurs at the site of receptor–ligand interaction [135,136]. Subsequently, arm-like membrane protrusions, known as pseudopods, then begin to protrude at the site of cytoskeletal disruption [137]. The formation of pseudopodia is universally driven by the nucleation of actin monomers via Arp2/3 activation to form and lengthen strands of F-actin. The induction of Arp2/3 in phagocytosis varies (Figure 6A). For example, FcγR recognition of opsonins activates Arp2/3 via induction of WASP and N-WASP, while the phagocytic receptor recognition of C3a/C5a induces Arp2/3 via the Rho/Rac signaling cascade [129,138].

Upon reaching the target cell, the pseudopod then diverts at the base of the target cell into two pseudopodia, leading to the formation of the nascent end of an invagination of the phagocyte cell membrane, known as the phagocytic cup (Figure 6B) [139,140]. Further development of the phagocytic cup occurs as F-actin and pseudopodia form adjacent to the side of the target cell as a result of actin nucleation via sustained Arp2/3 signaling and lengthening by myosin II motor proteins (Figure 6C) [141,142]. Concurrent with pseudopodia extension, an F-actin also forms perpendicular to the base of the phagocytic cup (Figure 6D) [142]. This contractile ring then migrates from the nascent to cortical end of the phagocytic cup as pseudopodia lengthening occurs, squeezing out any extra-particle fluids and ensuring that the pseudopodia conform to the surface of the target cell [143]. The contractility of the ring is primarily driven by the enzyme myosin light-chain kinase, which powers the myosin II and IXb motor proteins to continually contract and relax the ring (Figure 6E) [141,142,144]. The transition from the phagocytic cup to a complete phagosome occurs when the cortical end of the target cell becomes fully engulfed upon the successful fusion of the pseudopodia alongside the polymerization of F-actin via increased PI3K signaling (Figure 6F) [145]. The phagosome is then endocytosed by the phagocyte and fused to endosomes and lysosomes to undergo lysis [146].

#### 3.5.3. Conservation Between the Mechanism of Phagocytic Cup Formation and Other Methods of Cell–Cell Trafficking

As described in the above sections, the signaling cascade pathways described in each cell–cell trafficking mechanism all rely on actin polymerization to form pseudopodia or invadopodia, with the exception of cell fusion. There are some differences in upstream inducers of each cell–cell trafficking mechanism, such as the TNFAIP2-mediated formation of TNTs and TMs [122]. However, activation of these pathways ultimately feeds into some form of Cdc42/Rho/Rac-mediated nucleation of actin monomers and subsequent F-actin polymerization. While some clear differences exist between mechanisms—such as the requirement for full cell–cell contact in entosis versus only proximity of two cells for TNT and TM formation—these distinctions become less clear when comparing mechanisms such as trogocytosis and entosis.

Some researchers believe that trogocytosis is the result of an incomplete phagocytic cup formation. Krendel & Gauthier visualized the “wringing” of a target particle driven by many of the same myosin motors found on the F-actin ring that initially forms at the base of a phagocytic cup and defined this partial transfer as trogocytosis [140]. The contractile forces generated by this ring are hypothesized to have partially sheared the target particle due to an excess of protrusive and contractile forces. These contractile forces were also paired with the early termination of filopodia formation, essentially resulting in an early termination or closure of what would normally be a phagosome (Figure 6G) [140]. However, it is important to note that the definition of trogocytosis is specifically the partial transfer of membrane fragments or membrane proteins. Krendel & Gauthier only observed this interaction while using a fluorescent PAGE bead as a surrogate for the donor cell, which ultimately can only determine that there was the partial transfer of a particle rather than transfer of the membrane or functional membrane proteins [140]. Until this mechanism is shown between two living cells, it cannot be construed as the definitive mechanism for trogocytosis.

## 4. Outcomes of Cell–Cell Trafficking Events in the Context of Cancer

### 4.1. Trogocytosis

#### 4.1.1. Reprogramming Antitumor Immunity to Alter Tumor Survivability

Trogocytic interactions within the TIME can alter tumor cell survival and influence transcriptomic changes. Poupot et al. were the first to observe the lymphocyte trogocytosis of tumor cells in 2005 and concluded that immune cells, specifically γδ-T cells, could trogocytose anaplastic lymphoma cells in vitro [147]. This interaction also resulted in a subset of memory γδ-T cells that could bind to and trogocytose anaplastic lymphoma cells, as well as in increased sensitization of cytotoxic γδ-T cells [147]. Most notably, trogocytosis did not occur between cocultured allogeneic PBMCs from two different patients, thus highlighting how the immune cell trogocytosis of tumor cells could increase immune cell function [147]. Trogocytosis may also lead to a direct increase in the cytotoxic capabilities of the trogocyte. For example, NK cells displayed an increased trogocytosis of HER2 from trastuzumab-treated HER2+ breast cancer cells [22]. Acquisition of HER2 by NK cells resulted in increased expression of CD107a/LAMP1, resulting in increased cytoxic function of NK cells [22]. In addition, NK cells may also obtain TYRO3 via trogocytosis to achieve the same effect [148].

In contrast, trogocytosis can also decrease antitumor immunity. Hasim et al. reported in 2022 that NK cells could exogenously express functional PD1 protein after undergoing trogocytosis with double-positive PD1^+^/PD-L1^+^ leukemia cells [18]. Tumor cells can also obtain the immunoregulatory proteins CTLA4 and Tim3 from immune cells [17]. In both of these cases, the transferred proteins were still functional post-trogocytosis and led to a dampened immune response in vivo [17,18]. The transfer of genomic DNA has also been attributed to the process of trogocytosis, though it is currently unknown whether this transfer occurs as a consequence of trogocytosis or is a separate trafficking mechanism that occurs concomitantly [17]. The trogocytosis of tumor-specific antigens (TSAs) may also lead to increased immune cell fratricide [19,20,21]. In an in vitro and in vivo model of melanoma, T cells obtained Class I peptide-MHC (pMHC) from conventional dendritic cells (cDCs) containing a neoantigen produced by melanoma cells [19]. Lymphoid trogocytosis of cDCs led to increased T cell fratricide by CD8+ T cells primed against the trogocytosed pMHC complex, leading to decreased immune response [19].

#### 4.1.2. Trogocytosis-Mediated Impediment of Recombinant Therapeutics

CAR-T cells, a type of T cell with T cell receptors that were engineered ex vivo to detect specific target antigens expressed on tumor cells, are a promising novel immunotherapy [149]. However, despite an initial increase in tumor cell death, recurrence is common post-CAR-T cell immunotherapy and stems from the reduced expression of the neoantigen recognized by the CAR [149]. Although the loss of the target antigen is often permanent in recurring tumors, CAR-T cell trogocytosis of tumor cells contributes to a mutable reduction in target antigen expression, resulting in increased CAR-T cell fratricide and decreased antitumor immunity [149,150]. In the Pagliano et al. study, target antigen expression by cancer cells was not considered. A reduction in one specific TSA may not be expected, as conventional T cells can be stimulated by a multitude of antigen peptides [19,149,150]. In contrast, CAR-T cells are typically primed against a smaller subset of neoantigens, which may also fail to account for the heterogeneity of the tumor, suggesting that significantly reduced antigen abundance on tumor cells occurs primarily during CAR-T cell trogocytosis [151].

Both conventional and CAR-T cells undergo trogocytosis during TCR/CAR recognition of a neoantigen by an antigen-presenting cell (APC), indicating that trogocytosis is associated with an inherent process required for T cell activation [45]. Zhou et al. were able to attenuate CAR-T cell trogocytosis in 2023 by fusing the C terminal domain of CTLA4 to the cytoplasmic tail of the CAR. Endocytosis of CTLA4 is an intrinsic and continuous function of a T cell and leads to the rapid degradation and recycling of CTLA4 protein [45]. By fusing the C terminal domain of the CAR to CTLA4, the CAR is endocytosed concomitantly with CTLA4, leading to increased recycling of both receptors [45]. This decreased binding of CAR to a ligand led to decreased trogocytosis and less T cell fratricide [45].

In contrast to the findings regarding immunomodulation, a mechanism by which T cell trogocytosis induces the stimulation of adaptive immunity has recently been characterized [152]. CD28 is a receptor protein that recognizes B7 ligands (CD80/86) expressed on APCs [153]. Costimulatory signaling of CD28-B7 is dependent on TCR recognition of MHC peptides [153]. In an article authored in 2023, Xu et al. outlines a mechanism by which T cells can trogocytose CD80/86 ligands in a CD28-dependent fashion [152]. This interaction did lead to the depletion of CD80/86 expression on the APC, which decreased the activation of naive T cells; however, increased autostimulation was observed in T cells [152]. CD28-mediated trogocytosis also occurs independently of TCR-recognition of MHC peptides [152]. Although this study did not investigate trogocytosis-induced autostimulation within the context of cancer, it indicates that CD28 trogocytosis of CD80/86 allows for the clonal expansion of specific T cell populations with TCRs primed against a potent neoantigen, which could lead to increased tumor clearance. Reduced availability of CD80/86 on APCs would also decrease CTLA4-CD80/86 interactions, which leads to naive T cell exhaustion and the inactivation of the TCR. In contrast, increased autostimulation of a small population subset of T cells may also lead to decreased antitumor immunity, similar to the limitations in neoantigen recognition observed in CAR-T cells [45,149]. Supporting this hypothesis, blockade of CD28 or CTLA4 decreased regulatory T cell (T_Reg_) autostimulation, suggesting that antitumor immunity may be impeded as a result [152]. However, when combined with the therapeutic strategy of fusing the cytoplasmic domains of a CAR and CTLA4, harnessing CD28-mediated trogocytosis of CD80/86 may induce increased clonal expansion of CAR T cells effective against the tumor while decreasing CAR-mediated trogocytic acquisition of MHC peptides and thereby decreasing fratricide.

### 4.2. Entosis

The role of entosis in tumor development and progression has not been fully characterized. This is largely due to the multifaceted role of entosis in tumor evolution, with evidence that it can both promote and suppress malignancy. It is believed that entosis can act as a tumor suppressor by eliminating cancer cells that detach from the basement membrane [23,25,154]. At the same time, however, entosis can promote cancer progression by contributing to genomic instability and aneuploidy of host cells, supplying nutrients to malignant hosts and shielding invading cells from both immune surveillance and anti-cancer drugs [54,155,156,157,158]. Given that increased entosis correlates with increased disease stage and worse patient outcome, it is thought that entosis is generally beneficial for tumor growth [28,155]. Instances of a cell-in-cell phenotype have been observed in many different types of malignancies, including small cell lung carcinoma, ductal breast carcinoma, and other breast tumors, as well as renal cell carcinoma, colorectal cancer, and pancreatic ductal adenocarcinoma, to name a few [159,160,161,162,163,164]. Interestingly, though we will not be going into further detail here, entosis has also been implicated in the pathogenesis of non-cancerous diseases such as microcephaly [165].

Aneuploidy, otherwise known as an imbalanced number of whole chromosomes or chromosomal arms, is known to be a major driver of tumor development and progression [166,167]. Entosis is known to contribute to aneuploidy in human breast tumors by disrupting normal cytokinesis of the host cell [155]. This can occur when an engulfed cell blocks the cleavage furrow of its dividing host. Instead of undergoing normal cell division, it is possible for the host cell to integrate the internalized cell’s nucleus, resulting in a binucleated cell [155]. If the binucleated cell attempts to undergo division again, it often promotes aneuploidy through uneven chromosome distribution and breakage in the daughter cells [168]. In normal cells, p53 and other tumor suppressors help maintain the stability of the genome and prevent aberrant chromosome numbers [169]. However, in cancer, it is common for these critical regulators to be lost, allowing aneuploidy caused by entosis to perpetuate and contribute to more severe malignancy [170].

Approximately 50% of engulfed cells undergo lysosomal degradation by their host, allowing the engulfing cell to scavenge nutrients such as glucose and amino acids from the corpse of the loser cell [25,54,55]. It is believed that this may be a mechanism by which cancer cells can persist in a high-stress, low-nutrient setting such as the TME. In fact, Hamann et al. have shown that when placed in starvation conditions, MCF-7 cells that ingest their neighbors proliferate 10-fold more frequently than their non-entotic counterparts [55]. In the context of tumor development, entosis could serve as a mechanism by which less-fit individuals are sacrificed to redistribute nutrients to the fittest cells, thus ensuring the survival of the cancer population. This suggests that cells may have some ability to sense fitness through points of contact such as adherens junctions, which remains an important topic for further studies.

One of the more novel ways in which entosis contributes to tumor progression is through promoting drug resistance. Drug resistance remains a significant limiting factor when it comes to achieving cures for cancer patients. Although many types of cancers are initially sensitive to certain anti-cancer drugs and therapies, as the disease progresses, they often acquire resistance to these interventions and become more severe [171]. Recent studies have shown that entosis may contribute to therapeutic resistance by enabling cancer cells to evade anti-cancer therapies by “hiding” inside other cells [52,158,172]. Several different anti-cancer drugs have been implicated in causing the formation of CICs, including FOLFOX and FOLFRI regimens, taxanes such as Paclitaxel, and nintedanib. In addition to causing drug resistance, the induction of entosis through anti-cancer treatments has been shown to promote aneuploidy in drug-resistant populations [172].

### 4.3. Cell Fusion

Genomic instability is a characteristic of many types of cancers [173]. Cell fusion can contribute to this problem by causing polyploidy, or the addition of an extra set of chromosomes, in the final fused cell [60]. The new hybrid cell may acquire the biological characteristics of its parent cells or develop new ones, which can promote stronger colony formation, migration, proliferation, and survival abilities [174,175].

Cell fusion has also been shown to play an important role in facilitating metastasis to distant organs. Metastasis is a complex process that requires a cancer cell to overcome many barriers they do not typically face in the primary site, such as detaching from their point of origin, surviving in circulation, invading through layers of tissue, and surviving in an unfamiliar microenvironment [61]. Migratory bone marrow-derived cells, such as macrophages, have been shown to participate in cell fusion with tumor cells in mouse models of melanoma and breast cancer and are theorized to impart the characteristics needed to successfully metastasize on the resulting daughter cell [174,176]. It is also possible that fusion between a cancer and non-cancerous cell in the TME could result in a daughter cell that is better adapted for surviving in a new environment compared to its parent cells.

Cancer stem cells (CSCs) are a relatively rare subset of cancer cells that act as a reservoir of self-sustaining cells and are capable of self-renewing and maintaining the tumor [177]. CSCs have been identified in a wide range of both liquid and solid tumors, including multiple myeloma, pancreatic cancer, breast cancer, leukemia, and glioblastoma to name a few [178,179,180,181,182]. Several groups have suggested that cell fusion may be a mechanism by which CSCs are created, whether through fusion between somatic and stem cells or normal cells and tumor cells [183,184,185,186,187]. Cell fusion often results in daughter cells that are more tumorigenic than their parent cells, sometimes exhibiting CSC characteristics [183,188]. These fusion events could also help explain intratumoral heterogeneity. The daughter cell that results from cell fusion can have a unique combination of traits from its parent cells, which may result in a novel CSC phenotype that otherwise would not have arisen in the tumor.

Drug resistance is another significant challenge to overcome when treating cancer. Researchers have shown that cell fusion can contribute to drug resistance, either through the fusion of a non-resistant and resistant cell or even through fusion of non-resistant tumor cells with bone marrow-derived cells (BMDCs) [189]. Recently, it was even shown that the fusion between cancer cells and nearby muscle cells enhances the drug resistance of prostate tumors [64]. A possible mechanism for how cell fusion can contribute to drug resistance is through the formation of polyploid giant cancer cells when exposed to certain chemotherapies [190,191]. It has also been theorized that mitochondrial abnormalities resulting from cytoplasmic fusion may also contribute to drug resistance; however, more research is needed to confirm this notion [192,193].

### 4.4. Tunneling Nanotubes and Tumor Microtubes

#### 4.4.1. Transfer of Nucleic Acids

Within the context of cancers, TNTs and TMs are distinct from previous trafficking methods in that they only require two cells to be nearby and do not require physical contact [36]. However, the transfer of cellular components when facilitated by TNTs and TMs extends beyond membrane proteins and fragments, such as the acquisition of mitochondria and mtDNA [7,111,113]. Regulation of mtDNA is crucial for maintaining cellular respiration and metabolism. Through TNT- and TM-mediated shuttling, mitochondria and mtDNA can be transferred between two tumor cells, resulting in a tumor cell with increased proliferative capabilities. Additionally, tumor cells can also siphon mitochondria from other cells within the TME, such as mesenchymal stromal cells and platelets, leading to increased tolerance of reactive oxygen species (ROS), which induce DNA damage and cellular stress [194,195].

Tumor-infiltrating lymphocytes (TILs) are another population which can have their mitochondria and mtDNA hijacked by a tumor cell, resulting in decreased T cell oxygen consumption rate [7]. Fragmentation of mitochondria via antimycin A or rotenone did not decrease the rate of transfer, but did render any transferred fragments unviable [7,196]. Clinically, direct targeting of any aberrant mitochondrial fragments is not therapeutically possible, as the origins of the trafficked mitochondria would need to be ascertained and targeted against [7]. Additionally, any developed therapies would potentially need to be localized, as systemic therapies may ablate cells distal to the tumor, such as other mesenchymal stromal cells not located near the TME.

TNT-mediated trafficking of nucleic acids is not limited to mtDNA, as the transfer of microRNAs (miRNA), a type of non-coding RNA, is also observed to be trafficked by nanotubes. miR-19A is an oncogenic driver miRNA [197,198]. One of its oncogenic functions includes the negative modulation of MHC Class I gene expression, which has been implicated in antitumor immunity. Transfer of miR-19A was observed between two osteoblast cell lines, as well as between an osteoblast cell and a non-immortalized normal cell in vitro [198]. Other oncogenic miRNAs, such as miR-132 and miR-155, are also transferred via tumor TNT networks and result in the oncogenic transcriptional reprogramming of angiogenic and proliferation pathways [197,199]. Viral RNA can also be transferred by TMs, which contributes to the spread of viral pathogens. This has been observed across multiple species, including human PR8-influenza and human metapneumoviruses [200,201]. Although the transfer of cancer-causing viral RNA, such as the human papillomavirus and the Epstein–Barr virus, has not yet been implicated in malignancies, these observations suggest that replication and dispersion may also be facilitated by TNTs and TMs.

#### 4.4.2. Transfer of Functional Proteins

While much of the current focus of TNTs and TMs is on alteration of cellular respiration via the transfer of mitochondria, an often-underappreciated facet trafficking via TNTs and TMs is the transfer of functional oncogenic proteins. For example, Desir et al. observed TNT-mediated trafficking of mutant KRAS^G12D^ and KRAS^G13D^, two constitutively active variants commonly observed in pancreatic, lung, and colorectal cancers, using in vitro cocultures between two distinct colorectal cancer (CRC) cell lines [6]. Increased TNT formation was also associated with elevated KRAS^G12D/G13D^ expression [6]. Transfer of both KRAS^G12D/G13D^ mutants occurred when the mutation was endogenously expressed by the tumor cell, indicating that the formation and subsequent transfer via TNTs is an innate function of the tumor cell. Transferred KRAS^G12D/G13D^ was still functional and increased the expression of phosphorylated ERK (pERK), a downstream effector of activated KRAS. Increased TNT formation was observed in cells that received the mutant KRAS variants and also altered cell morphology, leading to decreased recipient cell size as early as 48 h post-coculture [6]. Taken altogether, these findings indicate that the expansion of a TNT network contributes to the formation of a heterogenous TME and may originate from a small number of oncogene-expressing tumor progenitor cells that then traffic oncogenic protein to induce aberrant signaling in recipient cells.

## 5. Discussion

While we have addressed each of these mechanisms of contact-dependent intracellular transfer separately for clarity in this review, it is important to note that they likely occur concurrently in human tumors. To date, trogocytosis, entosis, cell fusion, and tunneling nanotubes have largely been studied in a vacuum, with little research being conducted to identify how these processes influence and synergize with one another. However, future studies may reveal that these four mechanisms are more interconnected than previously believed. For example, oncogenic proteins such as KRAS can be transferred through TNTs [6]. These proteins can make the recipient cell more migratory and, as a result, may increase the relative rates of entosis and cell fusion, as both phenomena require some degree of cellular motility [6].

Furthermore, studies have shown that the membrane fragments transferred by trogocytosis have led to the formation of intracellular trogosomes. Although the assumption is made that this occurs due to the recipient cell “nibbling” the donor cell through actin protrusions, it may also be possible that residual protein and membrane fragments may be left behind in the host cell after the engulfed cell exits during entosis. This interplay may also link cell fusion and trogocytosis. During cell fusion, PS destabilizes the membrane between two cells, increasing membrane fluidity, which may increase the transfer of membrane fragments during trogocytosis [42]. Following the model proposed by Krendel & Gauthier, the contractile F-actin ring located on the phagocytic cup is hypothesized to “clamp” and “wring” the target cell or particle prematurely [140]. Although this would not normally occur due to the rigidity of an intact cell, cells with compromised membranes, such as ones observed during cell fusion, may lead to trogocytosis if the two fused cells are pulled away from each other prematurely.

To minimize drug resistance, immune evasion, and aberrant nutrient scavenging because of cancer cell entosis, one possible therapeutic strategy would be to target Rho kinase and the ROCK signaling pathway. As mentioned previously, this pathway is crucial for the invading cell to enter its host. Some clinical trials targeting Rho/ROCK signaling have seen some success in treating diseases such as glaucoma [202]. Other trafficking mechanisms such as TNTs and trogocytosis have also been attenuated using inhibitors that inhibit Rho/ROCK signaling. The use of these inhibitors has led to reduced trogocytosis-induced fratricide in T cells and CAR T cells both in vitro and in vivo. However, to date, no group has tested the efficacy of this strategy in preventing entosis in human patients.

Cell fusion is known to promote genomic instability, metabolic dysregulation, and contribute to the generation of cancer stem cells in tumor development. There are several key mechanisms that can be targeted to disrupt this process. For example, researchers have shown inhibition of syncytin-1, an important fusogen protein that drives cell fusion, limits the proliferation and metastasis of NSCLC in vitro [98]. Although these results do not directly address the role of syncytin-1 in promoting cell fusion, they demonstrate the viability of syncitin-1 as a therapeutic target in limiting cell fusion in human patients.

## 6. Conclusions

The contact-dependent processes of intracellular trafficking of materials such as trogocytosis, entosis, cell fusion, and tunneling nanotubes are starting to be recognized as important mechanisms by which cancer cells can mold the TIME to their advantage. These events can promote improved tumor cell immune evasion, drug resistance, intratumoral heterogeneity, metastasis, and many other pro-tumor processes that can lead to more severe disease outcomes. As the field of immunotherapy moves forward, it is critical that these contact-dependent trafficking events be researched in further detail.

The discovery of how trogocytosis can decrease the efficacy of CAR-T cell therapy or the ability of cancer cells to avoid anti-cancer drugs by hiding in their neighbors through entosis are just a few examples of how these processes will have serious implications for the next generation of immunotherapies.

Additionally, through its contributions to genomic instability, contact-dependent trafficking may play an unexpected role in initiating tumorigenesis. Further research is needed to fully elucidate the role these processes play, both in creating cancer cells and destroying them.

## Figures and Tables

**Figure 1 cancers-17-02268-f001:**
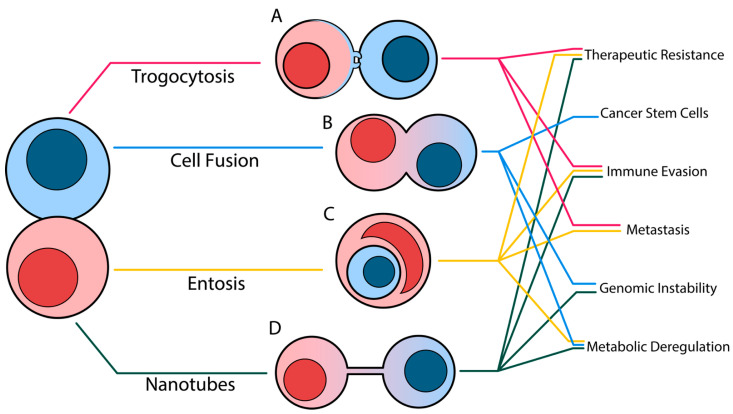
Graphical abstract depicting the four major contact-dependent cell trafficking processes (A) Trogocytosis, (B) Cell Fusion (C) Entosis, and (D) Tunneling Nanotubes and the potential ways they can influence tumorigenesis and progression. Lines indicate tumor-promoting characteristics that have been previously demonstrated for each process.

**Figure 2 cancers-17-02268-f002:**
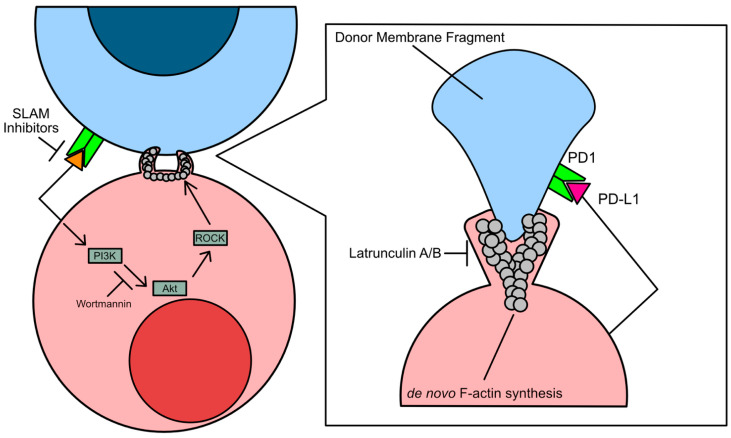
Overview of the mechanisms of trogocytosis. Trogocytosis is the transfer of a membrane fragment from a donor cell to a recipient cell. When these cells come into contact with each other, F-actin polymerization (grey circles) engages and results in the transfer of functional proteins from the donor cell, such as PD-1. This transfer can result in an immunomodulatory effect. Treatment with the PI3K inhibitor wortmannin or with the F-actin polymerization inhibitor Latrunculin A/B inhibits trogocytosis. Upstream inhibitors of PI3K-mediated ROCK F-actin polymerization, such as SLAM inhibitors, may also inhibit trogocytosis.

**Figure 3 cancers-17-02268-f003:**
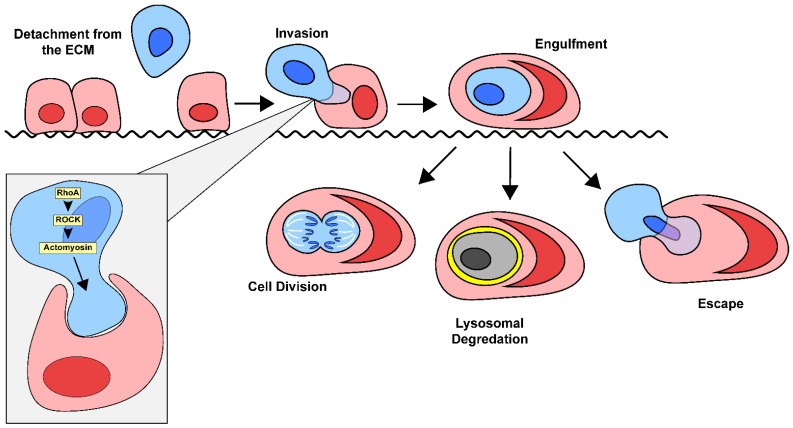
Overview of the mechanisms of entosis. Entosis is primarily initiated by the detachment of an adherent cell from the surrounding extracellular matrix. When the detached cell encounters an adherent neighbor, there is a difference in stiffness between them. The invading cell is typically stiffer than the adherent cell, which helps facilitate its engulfment. ROCK signaling in the invading cell results in mechanical tension, which promotes cellular invasion and entosis. After the loser cell is fully engulfed, it can undergo one of three possible outcomes. The invading cell may divide, be killed through lysosomal degradation, or escape its host cell.

**Figure 4 cancers-17-02268-f004:**
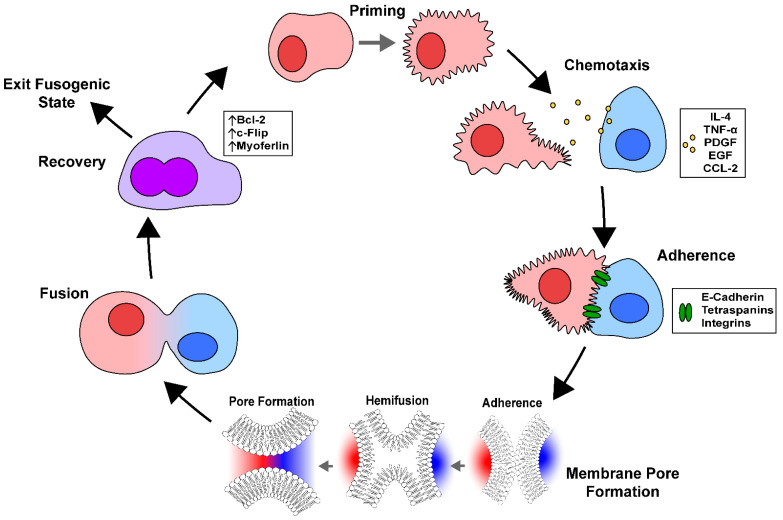
Overview of the mechanisms of cell fusion. Cell fusion initially begins with a priming phase, in which the fusing cells alter the architecture of their cell membrane by increasing surface membrane PS expression and prepare the machinery necessary for fusion to successfully occur. Next, there is a phase of chemotaxis, in which chemoattractant signaling between the two fusing cells promotes their migration towards each other. Some chemoattractants, such as CCL-2, have been involved in M2 macrophage recruitment to the site of the tumor. M2 macrophages and tumor cells have previously shown to undergo cell fusion, which results in increased tumor cell migration post-cell fusion. Once the cells are physically in contact, they adhere to one another using cell adhesion molecules, such as E-cadherin, tetraspanins, and integrins. After adherence, the cells begin to form a fusion pore. The most popular models suggest that as the cell membranes fuse, they form a hemifusion intermediate in which only the surface leaflets of the bilipid membranes are fused. Tension arising from this intermediate state drives fusion of the inner leaflets and leads to the formation of a membrane pore. This pore then expands to complete the fusion of the two cells into a single body. After fusion, the fused cell then undergoes a recovery phase in which the excess membrane is removed. After fusion, the cell may then exit its fusogenic state or repeat the process with another fusion partner.

**Figure 5 cancers-17-02268-f005:**
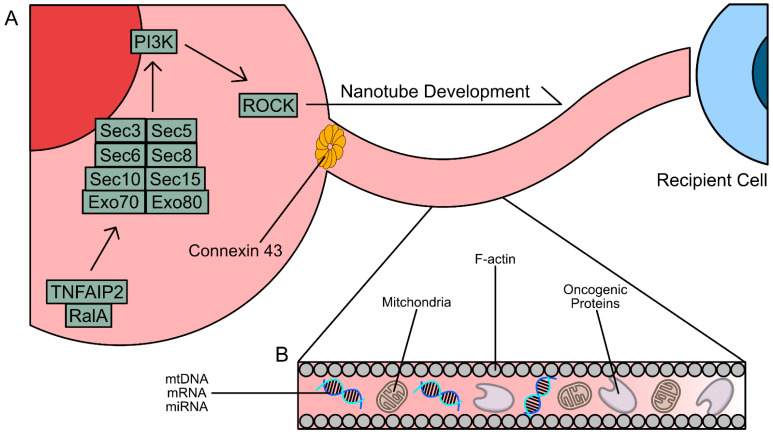
Overview of the mechanisms of TNT and TM formation. (**A**) TNTs and TMs form as a result of TNFAIP/RalA protein–protein interactions leading to the recruitment of the exocyst complex proteins. Downstream activation of PI3K and ROCK-mediated F-actin polymerization induce nanotube development. Connexin 43 gap junctions facilitate cargo entrance and exit at either end of the TNT or TM. The TNT also connects to the recipient cells via the connexin 43 gap junction and promotes F-actin formation by undergoing a process similar to the donor cell. (**B**) The cargo of TNTs and TMs can consist of nucleic acids, mitochondrial fragments, and functional proteins. Trafficked cargo can influence the gene and protein expression of the target cells.

**Figure 6 cancers-17-02268-f006:**
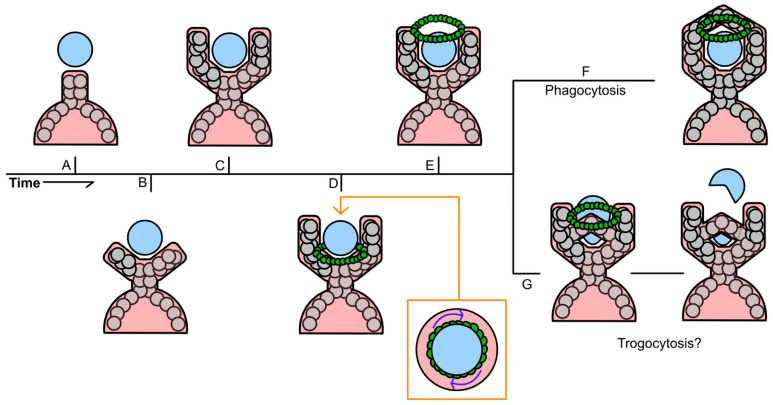
Overview of the mechanisms driving phagocytic cupping. Molecular steps of phagocytic cup formation. (**A**) Extension of F-actin (gray) filopodia (pink) to the base of the target cell or cell fragment (blue). (**B**,**C**) Divergence of the filopodia into two segments after reaching the base of the target cell. Each filopodia segment extends vertically upward the base of the target cell or particle, forming a bowl-like structure. (**D**,**E**) Formation of a contractile actin ring driven by myosin II and IxB (green). The contractile ring moves up along lateral scaffolding of the phagocytic cup. A top-down viewpoint schematic is illustrated within the orange box, and the direction of the contractile forces is illustrated with blue arrows. (**F**) Closure of the phagocytic cup to complete phagosome formation prior to internalization. (**G**) Illustration of a potential trogocytosis mechanism, where the closure of the phagocytic cup, seen in (**F**), occurs prematurely due to excess forces exerted by the contractile ring. The fragment captured by the incomplete phagocytic cup is then pulled away from the main body of the target cell or particle, as shown by the stretched blue oval in (**G**). In the case of phagocytosis, engulfment of the whole cell would be expected. In contrast, trogocytosis would only pinch off fragments of a cell or particle.

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
