# Peer review of "Molecular Dynamics of Trogocytosis and Other Contact-Dependent Cell Trafficking Mechanisms in Tumor Pathogenesis"

_cancers, 2025, doi:10.3390/cancers17142268_

Round 1
Reviewer 1 Report
Comments and Suggestions for Authors
Marcarian HQ et al well organized four recently characterized and underappreciated near-cell trafficking mechanisms: (i) trogocytosis, (ii) entosis, (iii) cell fusion, and (iv) tunneling nanotubes/tumor microtubes utilized by tumor cells to promote a hospitable microenvironment. However, some points should be revised.
It is recommended to stop using the microtubes and use only tumor microtubes (TMs). The authors defined TMs in sections 2.4 and 3.4.1 including their length and bore size. However, in section 4.4.1, the authors suddenly used the new term MTs twice perhaps this means microtubes and removed the term ‘tumor’ before microtubes. Also the term MTs is not listed in the abbreviations. The term microtubes is confusing as this term means Eppendorf tubes in biology. The abstract should also be corrected.
In Figure 2 left panel, two horns on the top of the bottom cell should be better drawn. These may be F-actin. F-actin should be depicted. Otherwise, left and right cartoons are too different.
Also, why the top part of the bottom cell is colored blue only in the left figure?
In Figure 3, the shapes of the red nucleus are crescent moon-like, which is different from Figure 1. Which are closer to true shapes?
What will happen after cell division in the cell-in-cell?
The cartoon of cell death appears apoptotic, but as written this should be lysosomal degradation or autophagy. The cartoon should be corrected.
Add the word PS in Figure 4.
In section 3.3.2, PDGF, TNFalpha, EGF, and IL-4 are listed as pro-chemotaxis factors. Are there any reports about chemokines promoting chemotaxis for cell adherence and fusion?
In Figure 5, is the outer part of F-actin surrounded by lipid bi-layered membrane? Or F-actin TNTs are not coated by membrane? Clarify this point.
In Figure 5, the TNT is not connected with the recipient cell. Add a description and figure about the mechanism of the connection of TNT with the recipient cell. E.g., Is this membrane fusion supported by adhesion molecules? Does the recipient cell provide actin?
Figure 6 should be the main figure as written in the title. There are many errors in the figure. Or the main title can be changed. What are gray and green balls? These must be written in the legend. Is the base of the target cell (blue) the same as the donor membrane fragment in figure 2? The shapes are different. In figure 2 The donor membrane fragment is pinched by F-actin horns, but in figure 6 the base of the target cell is not pinched.
In figure 2, two horns of F-actin are sticking out from the cell membrane, but in figure 6 the F-actins are surrounded by the cell membrane. Which are true? Correct the figures.
In the legend, “ divergence of the filopodia into two segments” is written. However, the shape can be dish-like if we see it in a 3D image/cartoon, but not two segments.
In section 4.4.1, line 672, microRNAs are not a type of long non-coding RNA. The sentence should be corrected.
Line 681, HMPV should be fully spelled out.
Abbreviations should be in alphabetical order.
Reviewer 2 Report
Comments and Suggestions for Authors
The paper, "Molecular Dynamics of Trogocytosis and Other Contact Dependent Cell Trafficking Methods in Tumor Pathogenesis," provides a comprehensive review of four unconventional near-cell trafficking mechanisms utilized by tumor cells: trogocytosis, entosis, cell fusion, and tunneling nanotubes/microtubes. The authors, Haley Q. Marcarian, et al, explore how these methods facilitate tumor cell proliferation and survival, alter anti-tumor immunity, and distribute oncogenic protein variants to foster a hospitable tumor microenvironment. This review article effectively synthesizes current knowledge on several underappreciated cell-cell trafficking mechanisms in the context of tumor pathogenesis. The paper is well-structured, providing historical overviews, detailed mechanisms, and implications for each trafficking method. The figures are helpful in illustrating complex processes.
Suggestions for Improvement
- Expanded Discussion on Therapeutic Implications: While the conclusion briefly mentions the critical need to research these events for future immunotherapies and the impact on CAR-T cell therapy, a more dedicated section or expanded discussion within each mechanism's section on current therapeutic strategies or potential drug targets related to these trafficking methods would enhance the paper's translational relevance.
- Clarification on "Near Cell Trafficking Mechanisms": The abstract states the review highlights "four recently characterized and underappreciated near cell trafficking mechanisms". While the contact-dependent nature is emphasized, a brief definition or elaboration on what "near cell" specifically implies in contrast to other forms of intercellular communication (e.g., distant metastasis or broader signaling pathways) might be helpful in the introduction.
- Discussion on Interplay between Mechanisms: The paper describes each mechanism individually. While distinct, there might be instances or conditions where these mechanisms can influence or overlap with one another. A brief discussion on the potential interplay, synergism, or hierarchical relationships between these various contact-dependent trafficking methods could offer a more integrated perspective.
- Future Directions and Unanswered Questions: While the conclusion touches on future research, expanding on specific, open questions for each mechanism (e.g., "it is currently unknown whether this transfer is a direct consequence of trogocytosis or driven by different interactions" ) within their respective sections or in a consolidated "Future Directions" section would highlight gaps in knowledge and stimulate further research.
A thorough proofreading for grammatical errors and awkward phrasing would enhance the overall readability of the manuscript. Such issues are common throughout the paper. A few examples are provided below:
Line 142-144: Studies assessing the 142immunomodulatory effects of trogocytosis have shown that Wortmannin and Latrunculin 143A/B by inhibiting F-actin synthesis
Incomplete sentence (major issue):
The sentence lacks a main verb or predicate for the clause after "have shown that..."→ What exactly have these studies shown about Wortmannin and Latrunculin A/B?
Misplaced modifier:
The phrase "by inhibiting F-actin synthesis" seems to be floating and not clearly tied to an action or outcome.
Parallel structure:
“Wortmannin and Latrunculin A/B” are both inhibitors, but it’s unclear what their shared effect is because of the incomplete sentence.
You could revise it like this:
"Studies assessing the immunomodulatory effects of trogocytosis have shown that Wortmannin and Latrunculin A/B suppress trogocytosis by inhibiting F-actin synthesis."
Line 162-166: “Should the mechanism be similar to immune cell trogocytosis of solid tumor cells, trogocytosisdriven by NK cells may be driven by SLAM receptor recruitment of PAK interacting protein (β-PIX), a guanine nucleotide exchange factor specific for downstream Rho/Rho-associated coiled-coil containing kinase (ROCK) mediated F-actin polymeriza-165tion”.
Typo / Missing space:
"trogocytosisdriven" should be "trogocytosis driven" (missing space).
Repetitive wording:
“Driven by” is repeated awkwardly: "trogocytosis driven by NK cells may be driven by..."This is redundant and affects clarity.
Clunky conditional clause:
“Should the mechanism be similar to immune cell trogocytosis of solid tumor cells” is grammatically correct (inverted conditional), but sounds stiff and may be better rewritten for clarity.
Unclear referent:
It’s a bit unclear whether “downstream Rho/ROCK-mediated F-actin polymerization” refers to the function of β-PIX or something else. The structure could be clarified.
Revised version for clarity and grammar:
"If the mechanism is similar to immune cell trogocytosis of solid tumor cells, NK cell–driven trogocytosis may involve SLAM receptor–mediated recruitment of PAK-interacting protein (β-PIX), a guanine nucleotide exchange factor that activates Rho/ROCK-mediated F-actin polymerization."
Line 189-190: “The invading cell may divide, be killed through lysosomal degradation, or it can escape its host cell.”
Parallelism issue: The first two parts (“may divide,” “be killed through lysosomal degradation”) follow the auxiliary "may", but the third part introduces a separate auxiliary “it can”, breaking the parallel structure.
Corrected version:
"The invading cell may divide, be killed through lysosomal degradation, or escape its host cell."
Line 192-193: “Loss of adhesion of cells from the extracellular matrix has been associated with entosis since the process was originally discovered.”
Wordiness:
"Loss of adhesion of cells from the extracellular matrix" is a bit awkward and wordy.
Clarity:
“Since the process was originally discovered” is correct but could be made more concise and smoother.
Improved version:
"Loss of cell adhesion to the extracellular matrix has been associated with entosis since its initial discovery."
Line 218-222: “Currently, it remains unknown whether non-transformed cells in vivo exhibit the same increase of entosis in response to UV radiation. This remains an important field of study as UV induced entosis or normal cells could play a role in early cancer cell transformation and genomic instability.”
"Increase of entosis" → should be "increase in entosis" (standard preposition usage).
"UV induced entosis or normal cells" → grammatically unclear. It likely means “UV-induced entosis of normal cells”.
Redundant wording: “Currently, it remains unknown...” could be simplified to “It remains unknown...” or “Currently, it is unknown...”
"Remains an important field of study" → could be made more concise and stronger.
Revised version:
"It remains unknown whether non-transformed cells in vivo exhibit a similar increase in entosis in response to UV radiation. This is an important area of investigation, as UV-induced entosis of normal cells may contribute to early cancer cell transformation and genomic instability."
Line 230-231: “However, when this process becomes dysregulated it can lead to the development of cancer.”
A comma is needed after the introductory clause.
Line 240-242: “Tension as a result of this intermediate state results in the fusion of the inner leaflets and the subsequent formation of a membrane pore.”
Redundant usage of “tension … results in … results in” – the phrase “results in” is repeated.
Wordiness – “as a result of this intermediate state” is lengthy and could be streamlined.
Revised version:
“Tension arising from this intermediate state drives fusion of the inner leaflets and leads to the formation of a membrane pore.”
Line 250-253: "During fertilization, it is believed that exposed phosphatidylserine (PS) localized to the anterior acrosomal region of sperm is critical for the membrane destabilization process that allows penetration of the zona pellucida."
Wordiness: "the membrane destabilization process that allows penetration" can be made more concise.
Punctuation: A comma before "localized" can help with readability, though it's optional.
Clarity: The sentence structure is a bit heavy and could be streamlined for easier comprehension.
Revised version:
"During fertilization, exposed phosphatidylserine (PS), localized to the anterior acrosomal region of the sperm, is thought to be critical for membrane destabilization that enables penetration of the zona pellucida."
Line 338-341: "TMs can oftentimes be visualized using brightfield or phase contrast microscopy whereas TNTs typically require higher fidelity imaging techniques such as scanning electron microscopy (SEM)"
Missing comma: A comma is needed after “microscopy” to separate the two clauses.
“Oftentimes” is slightly informal for scientific writing; “often” is preferred.
Revised version:
"TMs are often visualized using brightfield or phase contrast microscopy, whereas TNTs typically require higher-resolution imaging techniques, such as scanning electron microscopy (SEM)."
Line 355-356: "RalA interacts with guanosine-5’-triphosphate (B-GTP), a binding interaction catalyzed by 3-phosphoinositide-dependent kinase 1 (PDK1)"
Possible typo: "B-GTP" is not a standard abbreviation for guanosine-5'-triphosphate. It should be GTP (guanosine triphosphate). Or “RalA/B interacts with…”
Punctuation: The comma before "a binding interaction..." may be misleading. It suggests an appositive, but the structure is unclear.
Scientific accuracy:
The binding of RalA to GTP is typically mediated by guanine nucleotide exchange factors (GEFs), not PDK1 directly. PDK1 may regulate upstream components but doesn't catalyze GTP binding directly.
Line 371-373: "These signaling pathways also stimulate TNFAIP2 function, signifying that there is no one master regulator of TNT and TM formation as a potential therapeutic target."
Ambiguity in phrasing: The phrase "no one master regulator ... as a potential therapeutic target" is a bit awkward and could be misread.
Tone and precision: “Signifying” can be replaced with “suggesting” or “indicating” for better scientific tone.
Parallel structure: “TNT and TM formation as a potential therapeutic target” implies the formation is the target, not the regulator. That could be confusing.
Line 382-384: "TNTs and TMs are more prone to breakage resulting in the incomplete transfer of components between two cells when compared to transport with nanotubes shorter in length."
Punctuation: A comma is needed after "breakage" to separate the dependent clause.
Wordiness: “Nanotubes shorter in length” is unnecessarily wordy; “shorter nanotubes” is better.
“Transport with” can be more directly stated as “transfer through” or “transport via.”
Revised version:
"TNTs and TMs are more prone to breakage, resulting in incomplete transfer of components between cells compared to shorter nanotubes."
Line 388-389: “They are similar structurally and in terms of function when compared to TNTs” → refined to “the structural and functional similarities between TMs and TNTs” for clarity and flow.
Line 462-465: “While some obvious differences between mechanisms can be observed, such as the need for full cell-cell contact in entosis versus two cells only needing to be nearby for the formation of TNTs and TMs, those differences are increasingly unclear when comparing the mechanisms of trogocytosis and entosis, for example.”
Comma splice and punctuation — commas are needed to separate clauses clearly.
Awkward phrasing — the sentence is a bit long and complex, making it harder to follow.
Repetition of "differences" — can be rephrased to avoid redundancy.
Revised version: "While some clear differences exist between mechanisms—such as the requirement for full cell–cell contact in entosis versus only proximity of two cells for TNT and TM formation—these distinctions become less clear when comparing mechanisms like trogocytosis and entosis."
Line 466: “It is believed by some” is wordy and vague. Can be rephrased to "Some researchers believe”
Line 519-520: "Expression of the target antigen by the cancer cell was not accounted for in the Pagliano et al. study." Is unconcise, unclear and unformal, could be rephrased to “In the study by Pagliano et al., target antigen expression by the cancer cells was not considered.”
Line 602-604: Recently, researchers have shown that entosis can contribute to this problem by allowing cancer cells to shield themselves from anti-cancer therapies by “hiding” inside other cells.
Could be refined for conciseness, tone, and grammar as "Recent studies have shown that entosis may contribute to therapeutic resistance by enabling cancer cells to evade anti-cancer therapies through 'hiding' inside other cells."
Round 2
Reviewer 1 Report
Comments and Suggestions for Authors
2nd Peer-review of Trogocytosis review
The revised sentences according to my point out were not marked in yellow, but Marcarian HQ et al mostly revised the Ms according to my and reviewer #2’s suggestions. However, there are still some concerns.
- In the first peer review, I commented “In section 3.3.2, PDGF, TNFalpha, EGF, and IL-4 are listed as pro-chemotaxis factors. Are there any reports about chemokines promoting chemotaxis for cell adherence and fusion?” The response by authors is “ Thank you for bringing this to our attention. We have clarified the role of chemokines promoting chemotaxis for cell fusion.” and only one sentence “More research is needed to determine if these chemokines play 234 a role in promoting the fusion of cancer cells.” was added on the bottom of the section without adding references and without modifying Figure 4. By searching PubMed, some key references were found regarding chemokine and cell fusion as listed below. Additional sentences, citations, and Figure 4 modifications are needed.
- Xia C, Zhang Q, Pu Y, Hu Q, Wang Y. Cell fusion between tumor cells and macrophages promotes the metastasis of OSCC patient through the activation of the chemokine signaling pathway. Cancer Med. 2024 Feb;13(4):e6940. doi:10.1002/cam4.6940. PMID: 38457216; PMCID: PMC10923029.
- Ramakrishnan M, Mathur SR, Mukhopadhyay A. Fusion-derived epithelial cancer cells express hematopoietic markers and contribute to stem cell and migratory phenotype in ovarian carcinoma. Cancer Res. 2013 Sep 1;73(17):5360-70. doi:10.1158/0008-5472.CAN-13-0896. Epub 2013 Jul 15. PMID: 23856249.
- The word ‘Methods’ in the title sounds like experimental methods, so this should be replaced with another term such as events, phenomena, or mechanisms.
- The new discussion section is good but needs improvement in grammar to increase readability and understandability as reviewer #2 pointed out. Grammarly might help.
2nd Peer-review of Trogocytosis review
The revised sentences according to my point out were not marked in yellow, but Marcarian HQ et al mostly revised the Ms according to my and reviewer #2’s suggestions. However, there are still some concerns.
- In the first peer review, I commented “In section 3.3.2, PDGF, TNFalpha, EGF, and IL-4 are listed as pro-chemotaxis factors. Are there any reports about chemokines promoting chemotaxis for cell adherence and fusion?” The response by authors is “ Thank you for bringing this to our attention. We have clarified the role of chemokines promoting chemotaxis for cell fusion.” and only one sentence “More research is needed to determine if these chemokines play 234 a role in promoting the fusion of cancer cells.” was added on the bottom of the section without adding references and without modifying Figure 4. By searching PubMed, some key references were found regarding chemokine and cell fusion as listed below. Additional sentences, citations, and Figure 4 modifications are needed.
- Xia C, Zhang Q, Pu Y, Hu Q, Wang Y. Cell fusion between tumor cells and macrophages promotes the metastasis of OSCC patient through the activation of the chemokine signaling pathway. Cancer Med. 2024 Feb;13(4):e6940. doi:10.1002/cam4.6940. PMID: 38457216; PMCID: PMC10923029.
- Ramakrishnan M, Mathur SR, Mukhopadhyay A. Fusion-derived epithelial cancer cells express hematopoietic markers and contribute to stem cell and migratory phenotype in ovarian carcinoma. Cancer Res. 2013 Sep 1;73(17):5360-70. doi:10.1158/0008-5472.CAN-13-0896. Epub 2013 Jul 15. PMID: 23856249.
- The word ‘Methods’ in the title sounds like experimental methods, so this should be replaced with another term such as events, phenomena, or mechanisms.
- The new discussion section is good but needs improvement in grammar to increase readability and understandability as reviewer #2 pointed out. Grammarly might help.
Author Response
Please See Attached for our line by line response

Reviewer 2 Report
Comments and Suggestions for Authors
The authors have adequately addressed nearly all of the reviewer’s comments and made substantial improvements to the manuscript. The revisions have significantly enhanced the clarity and overall quality of the work.
I have no further major concerns.
Round 3
Reviewer 1 Report
Comments and Suggestions for Authors
Line 245 should be corrected.
Author Response
Thank you for your comment and your time - We have revised line 245 to "Cell fusion can also result in a more metastatic and mobile phenotype of the cancer cell." as that was our intended point of the statement. Thank you for pointing this out to us.